# Targeting Mitochondrial DNA Transcription by POLRMT Inhibition or Depletion as a Potential Strategy for Cancer Treatment

**DOI:** 10.3390/biomedicines11061598

**Published:** 2023-05-31

**Authors:** Sabrina C. D. Daglish, Emily M. J. Fennell, Lee M. Graves

**Affiliations:** Department of Pharmacology and the Lineberger Comprehensive Cancer Center, University of North Carolina at Chapel Hill, Chapel Hill, NC 27599, USA; sdaglish@unc.edu (S.C.D.D.); emily_fennell@med.unc.edu (E.M.J.F.)

**Keywords:** mitochondria, mitochondrial transcription, oxidative phosphorylation, POLRMT, ClpP, mitochondrial DNA, mechanism of action

## Abstract

Transcription of the mitochondrial genome is essential for the maintenance of oxidative phosphorylation (OXPHOS) and other functions directly related to this unique genome. Considerable evidence suggests that mitochondrial transcription is dysregulated in cancer and cancer metastasis and contributes significantly to cancer cell metabolism. Recently, inhibitors of the mitochondrial DNA-dependent RNA polymerase (POLRMT) were identified as potentially attractive new anti-cancer compounds. These molecules (IMT1, IMT1B) inactivate cancer cell metabolism through reduced transcription of mitochondrially-encoded OXPHOS subunits such as ND1-5 (Complex I) and COI-IV (Complex IV). Studies from our lab have discovered small molecule regulators of the mitochondrial matrix caseinolytic protease (ClpP) as probable inhibitors of mitochondrial transcription. These compounds activate ClpP proteolysis and lead to the rapid depletion of POLRMT and other matrix proteins, resulting in inhibition of mitochondrial transcription and growth arrest. Herein we present a comparison of POLRMT inhibition and ClpP activation, both conceptually and experimentally, and evaluate the results of these treatments on mitochondrial transcription, inhibition of OXPHOS, and ultimately cancer cell growth. We discuss the potential for targeting mitochondrial transcription as a cancer cell vulnerability.

## 1. Mitochondrial Transcription and Metabolism as Targets for New Anti-Cancer Approaches

There is growing interest in the role of mitochondria in cancer due to their involvement in cellular metabolism, apoptotic regulation, redox balance maintenance, and activation of the integrated stress response and related immune responses [1,2,3,4,5,6,7]. Mitochondria are unique in that they have their own DNA (mtDNA) which must be transcribed and translated to make RNAs and proteins necessary for mitochondrial function [8]. The mitochondrial genome is composed of circular double stranded DNA containing a light and heavy strand that encodes genes for 2 ribosomal RNAs, 22 transfer RNAs, and 13 proteins essential for oxidative phosphorylation (OXPHOS) [8]. Transcription of these genes is mediated by the light strand promotor (LSP) and two heavy strand promotors (HSP1, HSP2) [9]. mtDNA-encoded genes are involved in mitochondrial translation, OXPHOS, neuroprotection, and gene expression regulation [8,10,11,12]. There are numerous pharmacological tools that target these mitochondrial processes, including electron transport chain inhibitors (i.e., metformin, rotenone, and oligomycin), TCA cycle inhibitors (i.e., CPI-613 and AGI-519), and antibiotics that inhibit mitochondrial translation (i.e., doxycycline and tetracycline) [13,14,15,16,17]. Some of these compounds have shown clinical potential for the treatment of cancer and other diseases, and new ways to target mitochondria are continually being discovered [7,18].

Inhibition of mitochondrial transcription has recently been pursued as a novel anti-cancer approach due to the significant effects observed on cancer cell viability [19,20]. While little is known about the direct importance of mitochondrial transcription in cancer, a number of studies have examined the relationship between mtDNA copy number and cancer [21,22,23,24], and dysregulation of mitochondrial transcription has been linked to many of the “hallmarks” of cancer [25,26,27]. However, it is currently unclear whether copy number changes in cancer are due to rapid division or selection for advantageous mutations, since some cancers display increases or decreases in mtDNA copy number compared to normal cells [21,28,29]. Many mtDNA mutations have been discovered in cancer, but it is unclear if these mutations are beneficial to the cancer cells [24,30,31]. Regardless, cancer-dependent differences in mtDNA copy number or mitochondrial transcription could provide a unique opportunity to selectively arrest cancer cells while avoiding normal cells. In support of this, small molecules targeting the mitochondrial DNA-dependent RNA polymerase (POLRMT) have shown promise in pre-clinical studies. These compounds (IMT1, IMT1B) appear to inhibit cell proliferation by disrupting mitochondrial function and inducing an ATP-depleted energy crisis in breast and ovarian cancer [20,32].

Another evolving approach to targeting mitochondrial transcription includes the recently discovered small molecule activators of the mitochondrial protease ClpP (herein referred to as ClpP agonists). These compounds include the imipridones ONC201 and ONC206 (currently in clinical trials), the more potent TR compounds, and other recently discovered molecules [33,34,35,36,37]. Studies from our laboratory and others show that ClpP agonists rapidly deplete POLRMT protein and other key proteins involved in mtDNA transcription [33,34,35,38]. Despite an incomplete understanding of the mechanism of action, ClpP agonists are currently under investigation as potential anti-cancer compounds due to their significant growth inhibitory effects on many different cancer cell lines and cancer models [35,39,40,41,42].

Given the emergence of these potential strategies for targeting mtDNA or mitochondrial transcription in cancer, the objective of this paper is to summarize current studies exploring these approaches. We specifically compare and contrast known results of both POLRMT inhibition and depletion and provide initial data comparing these interventions in a common cell model. Lastly, we discuss the promise of inhibiting mitochondrial transcription as a cancer cell vulnerability with an emphasis on identifying promising new therapies for the treatment of various cancers.

## 2. Functional Roles of POLRMT in Mitochondria

POLRMT, the only RNA polymerase present in the mitochondria, transcribes and facilitates the replication of mtDNA [43]. POLRMT is therefore a pivotal enzyme for the expression of the genes encoded by mtDNA and for the maintenance of functional mitochondria and related metabolic processes [44]. During mitochondrial transcription, POLRMT works in concert with mitochondrial transcriptional factor A (TFAM) and mitochondrial dimethyladenosine transferase 2 (TFB2M) to initiate mitochondrial transcription. TFAM binds to mtDNA promoters to direct POLRMT to initiate transcription, while TFB2M aids POLRMT in melting and stabilizing the mtDNA strands to enable transcription (Figure 1) [9,43]. TFAM has an additional role in mtDNA maintenance. TFAM binds nonspecifically to mtDNA, causing the DNA to bend and allowing the packaging of genetic material into nucleoids [9,45]. TFAM typically binds to mtDNA every 15–20 kb, and the abundance of TFAM bound to mtDNA is critically important in mtDNA transcriptional regulation [46,47]. Overexpression of TFAM has been shown to reduce transcription and expression of mitochondrially-encoded genes, potentially due to TFAM blocking POLRMT access to mtDNA [47,48]. However, some tissues have been observed to upregulate expression of POLRMT, potentially as a mechanism to overcome high TFAM levels and promote increased transcription [47]. For example, transcription at LSP, HSP1, or HSP2 can be regulated by the TFAM protein level because each promotor is transcribed optimally at a different TFAM:mtDNA ratio [47,48]. These data suggest both TFAM and POLRMT play critical roles in regulating mitochondrial transcription and function.

When mitochondrial transcription is initiated from the LSP, POLRMT can also function as a primase (Figure 1) [49]. It is hypothesized that during mtDNA transcription, a G-quadruplex forms between RNA and DNA that causes POLRMT to stall, which instead allows for mtDNA replication to occur [50]. It is possible that the mitochondrial transcription elongation factor (TEFM) is involved in preventing the G quadruplex from forming, preventing replication and allowing transcription to occur instead [49,51,52]. However, if POLRMT stalling does occur, mtDNA replication is initiated and DNA polymerase **γ** (POLG) and other replication factors are recruited (mitochondrial single-stranded DNA-binding protein 1 (SSBP1), Twinkle mtDNA helicase (TWNK), etc.) to replicate the mtDNA (Figure 1) [8,51]. Mitochondria contain multiple copies of mtDNA and this number can vary greatly between cancer types [21]. Since mtDNA is less protected from damage and mutagenesis than nuclear DNA, additional copies may be required to maintain a heteroplasmic state where deleterious mutations are not pathological [29]. For an in-depth review of the effects of mtDNA damage on cellular function, see Nadalutti et al. [53].

## 3. POLRMT Inhibition as an Anti-Cancer Strategy

POLRMT is overexpressed at both the mRNA and protein level in several cancers, including breast, skin, lung, endometrial cancer, and osteosarcomas [32,54,55,56,57,58]. In both primary tumors and cancer cell lines, POLRMT overexpression promoted cell proliferation, migration, and invasion [32,54,55,56,57,58]. Knockout of POLRMT also impaired multiple cancer-related processes such as cell proliferation, migration, invasion, and angiogenesis [32,54,56,57,58]. Additional studies found that depletion of POLRMT induced apoptosis, as measured by caspase-3 and PARP cleavage, and increased TUNEL staining [56,57]. Mitochondrial membrane depolarization was also observed following POLRMT knockout, indicating that POLRMT activity is essential for mitochondrial function [57]. Despite this knowledge, the potential for targeting POLRMT and associated processes in cancer has only recently been explored.

Small molecule antiviral ribonucleosides (AVRNs) were first introduced in clinical trials for Hepatitis C as a strategy to inhibit the HCVRNA-dependent RNA polymerase [59]. An off-target effect of AVRNs was inhibition of POLRMT, leading to increased interest in identifying POLRMT inhibitors [60]. To screen for novel inhibitors of POLRMT, Bergbrede et al. developed a cell-free assay which identified the small molecule IMT1 as an inhibitor of POLRMT and mitochondrial transcription [19]. Further characterization determined that IMT1 selectively binds and inhibits POLRMT over other RNA polymerases found in humans, yeast, bacteria, and viruses [20]. IMT1 and its analogs (i.e., IMT1B), were shown to exert anti-cancer properties, most notably inhibiting cell proliferation and inducing cell death [20,32]. Consistent with POLRMT knockout experiments, IMT1-mediated inhibition of POLRMT impaired mitochondrial function through mtDNA depletion, OXPHOS inhibition [20], mitochondria depolarization, and mitochondrial reactive oxygen species (ROS) level induction [32]. Since mtDNA encodes proteins essential for OXPHOS, inhibition of POLRMT and mitochondrial transcription was predicted to reduce respiration, which was confirmed in Bonekamp et al. (2020) [20]. While impairment of OXPHOS could explain the observed increase in mitochondrial depolarization and ROS levels, the exact mechanism by which IMT1-mediated OXPHOS impairment leads to inhibited cell proliferation remains to be established.

## 4. Small Molecule ClpP Agonists as Anti-Cancer Compounds

Recently, a novel class of mitochondria-targeting drugs were discovered based on their highly unusual property of activating the mitochondrial protease ClpP. For in-depth reviews on pharmacological ClpP activation see Wong and Houry (2019) and Wedam et al. [61,62]. ONC201 was first identified as a potential anti-cancer compound in a screen for small molecule inducers of tumor necrosis factor-related apoptosis-inducing ligand (TRAIL) [37]. ONC201 (Dordaviprone) and ONC206 have now progressed to phase I, II, and III clinical trials for gliomas, acute myeloid leukemia (AML), endometrial, breast, and other cancers [63,64,65]. More recently, Madera Therapeutics LLC developed scaffold analogs of ONC201 (TR compounds) which are highly potent and selective agonists of ClpP [33]. Studies confirmed that these compounds bind to ClpP, the proteolytic barrel of the ClpXP mitochondrial matrix protease. The ClpXP protease is comprised of two components, ClpP and the unfoldase ClpX, which plays an important role in protein quality control and mitochondrial proteostasis (Figure 2) [66,67]. Specifically, ONC201 and the TR compounds bind within hydrophobic pockets between ClpP subunits, the same groove in which the IGF loops of ClpX bind [35,66,68,69]. ClpP agonist binding results in axial pore widening and subsequent activation of ClpP in the absence of ClpX (Figure 2) [35,68]. Though ClpP agonists were found to inhibit cell proliferation in breast cancer and other cancer cell models, the anti-cancer mechanism of action is just beginning to be elucidated [33,35,39,42,70,71,72,73,74]. Studies have shown that ClpP activation induces mitochondrial stress and glycolytic dependence, inhibits OXPHOS, global protein synthesis, and cell proliferation, and dysregulates multiple mitochondrial-related events including TCA cycle function and heme biosynthesis [33,34,38,75]. Two of the most potent TR compounds, TR-57 and TR-107, were found to have significant effects on POLRMT [34,38], which will be discussed in more detail below.

Though ClpP activation results in the widespread loss of mitochondrial matrix proteins [38], identification of specific ClpP substrates is still being pursued. Recent studies identified multiple ClpP-interacting proteins and putative substrates through proximity ligation (Bio-ID) and N-terminome (HYTANE) profiling methods [35,76]. Proteomics profiling in conjunction with transcriptomics and metabolomics analyses has provided further insight into the cellular response to ClpP activation [38]. Ultimately, the determination of the protein substrates of activated ClpP will assist in a better understanding of the mechanism of action of ClpP-targeting compounds on cancer metabolism and growth.

POLRMT protein was shown to be strongly reduced in triple-negative breast cancer (TNBC) cells following ClpP activation by TR-57 and TR-107 [34,38]. Immunoblot and proteomic analysis revealed that POLRMT was significantly depleted after 24 h of treatment with TR-57 or TR-107, but was unaffected in ClpP knockout TNBC cell lines following treatment [34,38]. POLRMT subsequently was identified as a ClpP-interacting protein by Bio-ID, indicating that the loss of this protein may be due to direct degradation by ClpP [76]. TFAM, a known POLRMT interactor, was also strongly reduced at the protein level following ClpP activation in TNBC cells in a time- and dose-dependent manner [33,34,75]. TFAM was also identified as a potential ClpP interactor by Bio-ID [76], and determined to be a direct ClpP substrate in response to activation by TR-65 [35]. TEFM, the elongation factor for mitochondrial transcription, was similarly depleted after ClpP activation in TNBC [38]. Thus, the loss of these essential mitochondrial transcription proteins (POLRMT, TFAM, and TEFM) suggests that inhibition of mitochondrial transcription may potentially play an important role in the mechanism of action of ClpP agonists.

## 5. Mechanistic Similarities between ClpP Agonists and POLRMT Inhibitors

### 5.1. Inhibition of Cell Proliferation

ClpP agonists and POLRMT inhibitors have been shown to impair cellular proliferation in multiple cancer cell lines and models [20,32,33,71,77,78]. Some evidence suggests that ClpP agonists and POLRMT inhibitors can induce cell death, though these effects are generally minimal compared to the cytostatic effects observed on cell proliferation [20,32,74,75,77,79,80]. However, these results may depend on the specific ClpP agonist studied. For example, ONC201 treatment increased apoptosis in mantle cell lymphoma, AML, pancreatic ductal adenocarcinoma, multiple myeloma, and ovarian cancer cell lines, whereas there have been no reports of significant increases in apoptosis after TR compound treatment [77,79,80,81,82]. ONC201 was previously reported to be both a ClpP agonist and a dopamine receptor D2 antagonist, while the TR compounds are selective ClpP agonists [33,35,83], which may explain these differences. In a model of TNBC (MDA-MB-231 cells), Greer et al. (2018) observed cell death in ONC201-treated breast cancer cells, but observed mitochondrial membrane ballooning instead of blebbing which suggests a non-apoptotic mechanism of cell death [75]. Minimal effects on cell death were observed when ONC201 and the TR compounds were tested in MDA-MB-231 cells or an additional TNBC cell model (SUM159 cells) [34]. Thus, these studies in TNBC cells suggest that ClpP agonists exert their anti-cancer effects through cytostatic, rather than pro-apoptotic responses. Similar observations were obtained with IMT1, where inhibiting POLRMT activity in breast or endometrial cancer prevented cell proliferation and induced apoptosis in only a small percentage of the cell population [20,32]. It is not yet clear if these drugs reliably induce apoptosis in other cancer cells or if this is a common property of the anti-cancer response to ClpP agonists or POLRMT inhibitors.

### 5.2. Cytostatic to Cancer Cells, Harmless to Normal Cells

One of the unusual properties of ClpP agonists is that they appear to adversely affect cancer cells but have little or no effect on normal cells. Several groups have tested ClpP agonists on immortalized cell lines (i.e., HFF) or primary cells and observed no significant effects on cell proliferation, viability, or apoptosis as compared to the cancer cell models [42,68,75,84]. Additionally, ClpP agonists were reported to be well tolerated clinically, with few side effects observed in patients with refractory solid tumors and neuroendocrine tumors [64,65]. In contrast, other mitochondria-targeting drugs such as metformin and CPI-631 have shown negative side effects such as gastrointestinal tract disorders, nausea, and vomiting; however, these toxicities were not reported in ONC201 clinical trials [64,65,85,86]. Similarly, POLRMT inhibitors appear to be more effective at inhibiting cancer cell proliferation with minimal impact on normal cells [20,32,58]. IMT1 has been shown to inhibit cancer cell proliferation in endometrial, ovarian, cervical, and skin cancers, whereas primary endometrial, hematopoietic, and hepatocyte cells did not respond to IMT1 treatment [20,32,58]. To our knowledge, there are no published studies that have directly investigated why normal cells are resistant to ClpP agonists or POLRMT inhibitors, although it has been speculated that cancer cells may have a greater demand for mitochondrial transcription to support an increased demand for OXPHOS [26,27,31]. This prediction suggests that cancer cells more reliant on OXPHOS may respond better to IMT1, whereas normal cells that are not as reliant on OXPHOS may be less affected by POLRMT inhibition [32]. Understanding why these compounds do not impact normal cells to the same extent as tumorigenic cells may provide further insight into the mechanism by which these small molecules inhibit cancer cell proliferation and help identify future approaches to selectively target cancer.

### 5.3. Dysregulation of Cancer Cell Metabolic Programs

Both ClpP agonists and POLRMT inhibitors induce major metabolic alterations in cancer cells, most prominently observed with the inhibition of OXPHOS [20,34,75,78,83]. Both classes of compounds reduce the mitochondrially-encoded protein components required for OXPHOS [20,34]. However, the mechanism by which OXPHOS is impaired in response to ClpP activation is less well understood than the proposed mechanism of POLRMT inhibitors. Loss of POLRMT activity results in the direct inhibition of mitochondrial transcription, which in turn prevents the expression of the 13 protein-coding genes essential for respiration [20]. Significant decreases in mRNA levels of several of these genes have been shown following IMT1 treatment [20,32], confirming inhibition of mitochondrial transcription following POLRMT inhibition. This combined with the loss of their cognate proteins (i.e., COI) could explain the observed inhibition of OXPHOS in these cells [20]. Studies have shown that OXPHOS is also inhibited after ClpP activation in cancer cell lines [34,75,78], but have not determined the direct cause of inhibition. Since ClpP agonists trigger the loss of multiple mitochondrial proteins, including those required for mitochondrial transcription (i.e., POLRMT, TFAM, TEFM) and OXPHOS (i.e., NDUFS3, SDHA) [34,38], it can be proposed that ClpP activation is inhibiting OXPHOS either through direct degradation of OXPHOS proteins, or degradation of mitochondrial transcription proteins necessary for maintaining OXPHOS function.

Another common metabolic alteration observed following ClpP activation or POLRMT inhibition is AMPK activation [20,75]. Greer et al. (2018) observed a decline in ATP levels and an increase in AMPK phosphorylation (Thr172) following ONC201 treatment [75]. Similar effects were observed with IMT1 treatment [20]. AMPK activation is mediated by a high cellular AMP/ATP ratio [87], further indicating that inhibition of OXPHOS and/or other mitochondrial processes required for ATP generation is occurring in response to these two different mitochondrial targeting approaches.

As an expected consequence of OXPHOS inhibition, ClpP agonists have been shown to cause a compensatory increase in glycolysis [34,75,78]. A similar compensatory response has not been demonstrated following POLRMT inhibition, although it may be predicted. Bonekamp et al. (2020) reported a reduced maximal respiratory capacity following IMT1 treatment, but did not report data regarding the extracellular acidification rate, a proxy for glycolytic activity [20,88]. Cancer cells can adapt to use aerobic glycolysis (Warburg effect) as their primary energy source in place of aerobic respiration [89,90]. Both ClpP activation or POLRMT inhibition appears to increase glycolytic dependence in cancer cells (become more “Warburg-like”), although it is unclear why shifting cancer cells away from OXPHOS utilization would be therapeutically beneficial given that many cancers are already highly glycolytic [91,92]. That said, targeting mitochondrial transcription by either approach may provide a viable treatment option for cancer cells that have developed high OXPHOS rates and become resistant to chemotherapy [93,94].

These observations also suggest that cancer cells that are more reliant on OXPHOS for proliferation will respond better to ClpP agonists or POLRMT inhibitors. This includes cancer types and cancer populations known to be more OXPHOS-dependent, including cancer stem cells [95], chemotherapy-resistant cancer cells [93,94], hypoxic solid tumors [96], and many other cancer subtypes [97]. However, studies directly correlating OXPHOS dependence to sensitivity to ClpP agonists or POLRMT inhibitors remain to be completed. A recent study reported that breast cancer stem cells, a population that is commonly OXPHOS-reliant, demonstrated impaired mammosphere formation after ONC201, TR-65, and TR-57 treatment [78]. Additional studies need to be completed to determine if reliance on aerobic respiration is a key factor that influences the responsiveness of cancer cells to ClpP agonists or POLRMT inhibitors.

### 5.4. Loss of mtDNA Content

An important similarity between ClpP agonists and POLRMT inhibitors is the corresponding reduction in mtDNA content, despite observations that the mtDNA copy number appears to change at different rates. IMT1 treatment reportedly took ~96 h to reduce the mtDNA copy number to ~25% of its original number [20], whereas ONC201 treatment required less time (~48 h) to reduce the mtDNA copy number to the same level [75]. This may in part be determined by the rate of TFAM protein loss. TFAM expression is closely correlated to the mtDNA copy number, likely because it binds to and packages mtDNA [98]. ClpP activation results in a time- and dose-dependent decline in TFAM protein that correlates with observed decreases in mtDNA content [33,34,75,78], as do POLRMT inhibitors (Daglish, unpublished observations (IMT1)). As described above, TFAM was recently reported to be a ClpP substrate [35], indicating that a decrease in TFAM protein levels may occur by direct degradation in response to ClpP agonists. By contrast, TFAM degradation is unlikely to be a direct result of POLRMT inhibition as there is no evidence for ClpP activation after IMT1 treatment.

Importantly, another potential mechanism leading to a reduction in mtDNA is through loss of POLRMT protein or activity. POLRMT is critical to mtDNA replication as a primase [60], so without POLRMT activity, mtDNA cannot be replicated [49]. In response to POLRMT inhibition, mtDNA may become damaged and degraded without subsequent replication of new mtDNA, leading to an overall decline in mtDNA content [99]. Mechanistically, IMT1 inhibits POLRMT activity directly whereas ClpP agonists deplete POLRMT protein [34]. Therefore, a common response to both IMT1 and ClpP agonists is predicted to be loss of POLRMT activity. While this may potentially explain the decreased mtDNA copy number, the corresponding decrease in TFAM protein could also contribute to the reduction in mtDNA content after either ClpP agonist or POLRMT inhibitor treatments.

While it is logical that mtDNA depletion impairs OXPHOS due to loss of required mtDNA templates to produce OXPHOS subunits, it remains to be established whether depletion of mtDNA is required for the growth inhibition observed by these compounds [100]. To be relevant in a clinical context, it should be determined if mtDNA copy number changes can be demonstrated in clinical samples following these treatments. Several groups have observed decreased mtDNA content in cell and animal models after ClpP agonists and POLRMT inhibitors. However, a complete loss of mtDNA has not been demonstrated with either ClpP agonist or POLRMT inhibitor treatment, suggesting that recovery of the mtDNA copy number may be possible [20,34,75]. While cells without mtDNA (p^0^ cells) are resistant to ONC201 [75], the effects of POLRMT inhibitors on p^0^ cells have not been reported. Comparing p^0^ cells to ClpP agonist- or POLRMT inhibitor-treated cells may help establish if the loss of mtDNA is a critical and common anti-cancer response to these treatments.

### 5.5. Inhibition of Mitochondrial Transcription

It has been shown that POLRMT and TFAM protein expression decreases after treatment with ClpP agonists, but it has not been established that mitochondrial transcription is inhibited. We predicted that mitochondrial transcription would be impaired by ClpP agonists due to their ability to deplete POLRMT, TFAM, and TEFM proteins which are all essential for mitochondrial transcription (Figure 1). Both TR-57 and IMT1 inhibited mitochondrial transcription in the tumorigenic immortalized HEK293T cells line (Figure 3) [101,102]. mRNA transcript levels of two mitochondrially-encoded protein genes required for OXPHOS (ND1 and ND6) were measured by quantitative PCR to monitor mitochondrial transcription from the heavy and light strands of mtDNA, respectively. Both treatments significantly decreased ND1 and ND6 transcript levels after 3 and 6 h, although there was a greater decrease observed following IMT1 treatment (Figure 3). This is potentially due to the different mechanisms by which IMT1 and TR-57 inhibit mitochondrial transcription. IMT1 directly inhibits POLRMT activity while TR-57 depletes POLRMT protein after 24 h [20,34]; therefore, TR-57 treatment may require longer treatment time to fully impact mitochondrial transcription. Additionally, it is not clear exactly how ClpP activation impacts mitochondrial transcription. Studies have shown a decrease of POLRMT, TEFM, and TFAM proteins after ClpP activation, and TFAM was recently determined to be a ClpP substrate [33,34,35,38,75]. It remains to be determined whether loss of one or all of these proteins is responsible for inhibition of mitochondrial transcription following ClpP activation. While our data support this hypothesis, further studies are needed to determine if inhibition of mitochondrial transcription is a common and important factor for the anti-proliferative effects of ClpP agonists or POLRMT inhibitors.

### 5.6. Summary of Similarities between ClpP Agonists and POLRMT Inhibitors

From the combined analysis of these studies, there are multiple similarities in the mechanism of action of ClpP agonists and POLRMT inhibitors. This includes depletion of mtDNA, inhibition of mitochondrial transcription, impairment of OXPHOS, activation of AMPK, metabolic remodeling, and ultimately, inhibition of cancer cell proliferation. From these observations, we propose that the similarities in mechanism between ClpP agonists and POLRMT inhibitors in part stems from the inhibition of mitochondrial transcription observed with both drug classes. While mitochondrial transcription may be impaired by different mechanisms, exploring the similarities and differences between ClpP agonists and POLRMT inhibitors may ultimately help understand the mechanism of action of both drug classes and evaluate the feasibility of targeting this mitochondrial process in cancer cells.

## 6. Mechanistic Differences between ClpP Agonists and POLRMT Inhibitors

### 6.1. Differences in Treatment Response Times

Despite the many mechanistic similarities between ClpP agonists and POLRMT inhibitors, there are several notable differences. Primarily, ClpP agonists inhibit cell proliferation much more robustly than POLRMT inhibitors. TR-57 strongly inhibits HEK293T cell proliferation following 72 h treatment, whereas IMT1 treatment requires a minimum of 120 h to show a similar inhibition of cell proliferation (Figure 4). In terms of potency, the TR compounds are more potent inhibitors of HEK293T growth; the IC_50_ of TR-57 in HEK293T cells is ~15 nM after 72 h, whereas the POLRMT inhibitor IMT1 has an IC_50_ of ~190 nM after 120 h of treatment (Figure 4). This difference in treatment response indicates that there are significant differences in the mechanism of action of these drug classes. Most notably, ClpP agonists are known to impact a number of metabolic pathways that are required for cancer cell proliferation (i.e., proline biosynthesis and heme biosynthesis pathways) [38,78].

Table 1 compares cell lines where IC_50_ data for both IMT1 and ClpP agonists has been obtained. Examination of this data shows that there is considerable variation in the growth inhibitory response depending on the cell and treatment type. Of the cell lines presented in Table 1, bladder, breast, ovary, and skin cancers responded better to ClpP agonists than IMT1, while colon, liver, lung, and prostate cancers responded better to IMT1. While Table 1 compares cell responses to ClpP agonists or POLRMT inhibitors individually, there are no known studies comparing the treatment responses of cancer cells to both ClpP agonists and POLRMT inhibitors. Table 1 also does not include the treatment times used in these dose-response experiments, due to the lack of reported information on this parameter. Importantly, our data in Figure 4 demonstrate a more substantial response (extent of growth inhibition, potency) to TR-57 compared to IMT1, suggesting that ClpP agonists may have advantages over POLRMT inhibitors. To our knowledge, the data shown in Figure 4 is the first attempt to directly compare the response to these two treatments in the same cell model.

### 6.2. Known and Unknown Mechanisms of Resistance

A remaining and important question is whether inhibition of POLRMT or activation of ClpP by small molecules results in the development of resistance in cancer cells. POLRMT inhibitors do not appear to be effective at inhibiting proliferation in all cancer cell lines in which they were tested. Table 1 highlights the cancer cell lines that did not respond to IMT1 treatment. By comparison, when the same cell lines were treated with ClpP agonists, their proliferation was clearly inhibited. For example, comparing responses in the breast cancer model MDA-MB-231 cells, IMT1 poorly inhibited growth (IC_50_ >30 μM), while all the ClpP agonists tested were highly effective at much lower concentrations (Table 1).

There are a few known cancer cell mechanisms of resistance to ClpP agonists, including p^0^ cells, fumarate hydratase knockout cell lines, and ClpP knockout or ClpP mutant cell lines [61]. The specific ClpP mutation, D190A, is known to confer resistance to ClpP agonists [68]. Comparatively, there are specific POLRMT mutations that confer resistance to IMT1. Bonekamp et al. (2020) reported six mutations in POLRMT that lead to IMT1-resistance [20]. A CRISPR screen aimed at identifying mechanisms of resistance against IMT1 found that loss of von Hippel-Landau protein (VHL) and mammalian target of rapamycin complex 1 (mTORC1) expression produced IMT1-resistance in RKO cells [104]. A similar CRISPR screen performed against ONC201 found that loss of ClpP or the mitochondrial intermediate peptidase (MIPEP) provided resistance to ONC201-mediated growth inhibition [105]. Since MIPEP is required to proteolytically activate ClpP, this could potentially explain how loss of this protein would contribute to the mechanism of resistance [105]. While both ClpP agonists and POLRMT inhibitors show promising anti-cancer properties, there is the potential for the development of resistance, though this may be less likely with ClpP agonists because they effect many pathways including the TCA cycle, protein synthesis, and proline biosynthesis [33,34,38,78,103].

### 6.3. Inhibiting One Protein versus Degrading Many Proteins

Major differences distinguishing the effects of POLRMT inhibitors from ClpP agonists may result from IMT1 selectively inhibiting mitochondrial transcription [20], whereas ClpP agonists dysregulate multiple mitochondrial processes [35,39,68,76,79]. As such, elucidating the specific mechanisms of action of ClpP agonists may prove to be more difficult. While efforts continue to identify specific ClpP substrates and related metabolic processes, the connection between loss of a specific ClpP substrate and its effects on cell proliferation remains to be established. Comparatively, a potential drawback to POLRMT inhibition is that these inhibitors only primarily impact one process or cancer cell vulnerability (mitochondrial transcription) [20]. Since pharmacological activation of ClpP proteolysis affects multiple growth-related cellular processes, this intuits an overall more disruptive mechanism than POLRMT inhibition. In the context of cancer cell growth, there may be an advantage to eliminating multiple events required for cancer cell growth to prevent the development of resistance more effectively. Figure 5 summarizes some of the potential differences and similarities between ClpP agonists and POLRMT inhibitors.

## 7. Summary

Targeting mitochondrial processes such as mitochondrial transcription is an emerging area of research. ClpP agonists and POLRMT inhibitors are promising new anti-cancer therapeutic strategies that may act through interrupting this important cellular function. Both ClpP agonists and POLRMT inhibitors inhibit mitochondrial transcription, deplete mtDNA, impair OXPHOS, and are potentially disruptive to multiple aspects of cancer cell metabolism, making them distinct from other mitochondrial targeted approaches (i.e., metformin, CPI-613, etc.) (Figure 6). Importantly, both ClpP agonists and POLRMT inhibitors are reported to inhibit cancer cell proliferation with minimal effects on normal cells [42,68,75,84]. Despite these commonalities, there are also important differences between ClpP agonists and POLRMT inhibitors. Preliminary evidence suggests there may be differences in how rapidly, potently, or broadly these compounds affect different cancer cell models. A survey of multiple studies indicates that ClpP agonists and POLRMT inhibitors may be differentially effective against different cancer types (Table 1). ONC201 and ONC206, having already advanced to phase I, II, and III clinical trials while the first-in-class POLRMT inhibitor, IMT1, is still in pre-clinical testing [32,63] suggests that there are advantages to ClpP agonists. Speculatively, due to their broad mechanism of action, ClpP agonists may have greater overall efficacy compared to drugs that only target mitochondrial transcription (i.e., POLRMT inhibitors).

Though ClpP agonists and POLRMT inhibitors effectively reduce cancer cell proliferation, much remains to be discovered about the mechanism of action of these compounds. An outstanding question is why ClpP agonists and POLRMT inhibitors are effective inhibitors of cancer cell growth, but not normal cell growth. As mentioned above, clinical trial data report ClpP agonists to be well tolerated with minimal side effects in cancer patients [64,65], further suggesting a minor impact on normal cells. Furthermore, as outlined in Table 1, there is much variation in drug response to IMT1 between cell lines as well as differences in response to ClpP agonists and IMT1 between cell lines. These differences in response have not been thoroughly investigated but may be important for understanding clinical applications of these therapies. Another major question is whether mtDNA loss is essential to the mechanism of action of these compounds, given that inhibition of mitochondrial transcription and depletion of mtDNA would induce functionally similar effects. Connecting the direct effects of inhibition of mitochondrial transcription and impairment of OXPHOS to the cause of cell growth inhibition needs to be further investigated.

Limitations to the current studies include a lack of studies directly comparing ClpP agonists and POLRMT inhibitors. While this review aimed to emphasize the surprising similarities between these anti-cancer approaches, how ClpP agonists and POLRMT inhibitors compare to other mitochondria-targeting drugs also remains to be determined. This is important because there are many cytotoxic effects to other mitochondria targeting drugs [106], so it is not yet clear if these compounds will induce similar negative side effects, or how they will respond in advanced clinical trials.

In summary, targeting mitochondrial transcription is a novel and potentially promising approach to disrupt cancer cell metabolism. Both ClpP agonists and POLRMT inhibitors have intriguing anti-cancer properties that will continue to be investigated as these compounds advance in clinical trials. In addition to emphasizing the importance of mitochondrial transcription in mitochondrial function, elucidating the mechanism of action of ClpP agonists and POLRMT inhibitors could lead to broader discoveries about cancer and how to better treat this prolific disease.

## Figures and Tables

**Figure 1 biomedicines-11-01598-f001:**
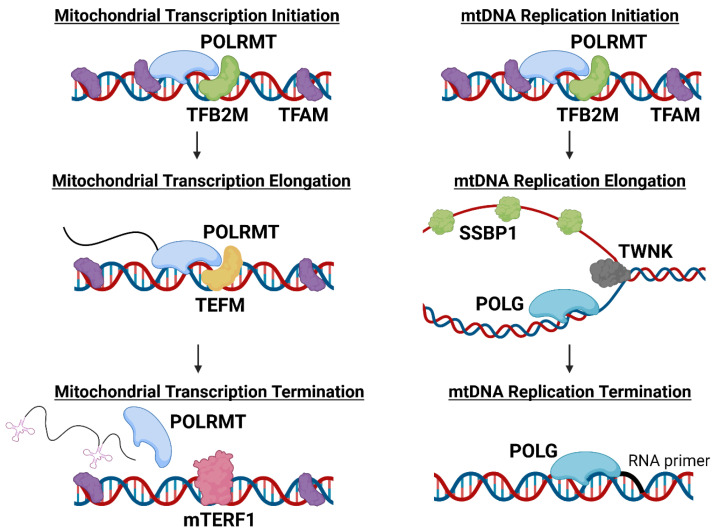
**The role of mitochondrial proteins in initiation, elongation, and termination of mitochondrial transcription and mtDNA replication.** The above schematic depicts mitochondrial transcription, mtDNA replication, and the respective role of mitochondrial DNA-dependent RNA polymerase (POLRMT) in each. Mitochondrial DNA (mtDNA) replication starts at the light strand promoter (LSP), whereas mitochondrial transcription can start from the LSP, the heavy strand promotor 1, or 2 (HSP1, HSP2). After mitochondrial transcription is initiated, mitochondrial transcription factor A (TFAM) and mitochondrial dimethyladenosine transferase 2 (TFB2M) dissociate from POLRMT and mitochondrial transcription elongation factor (TEFM) binds to stabilize POLRMT and mtDNA during elongation. After the polycistronic transcript has been created, POLRMT is sterically blocked by mitochondrial transcriptional termination factor 1 (mTERF1) (during LSP transcription), or by the 7S DNA region (during HSP2 transcription). Once POLRMT, TFAM, and TFB2M initiate mtDNA replication, the initiation complex dissociates and DNA polymerase γ (POLG) begins replication. Twinkle mtDNA helicase (TWNK) anneals the DNA strands while mitochondrial single-stranded DNA binding protein (SSBP1) binds to and stabilizes the single-stranded DNA strand. When POLG reaches the RNA primer again, replication is terminated.

**Figure 2 biomedicines-11-01598-f002:**
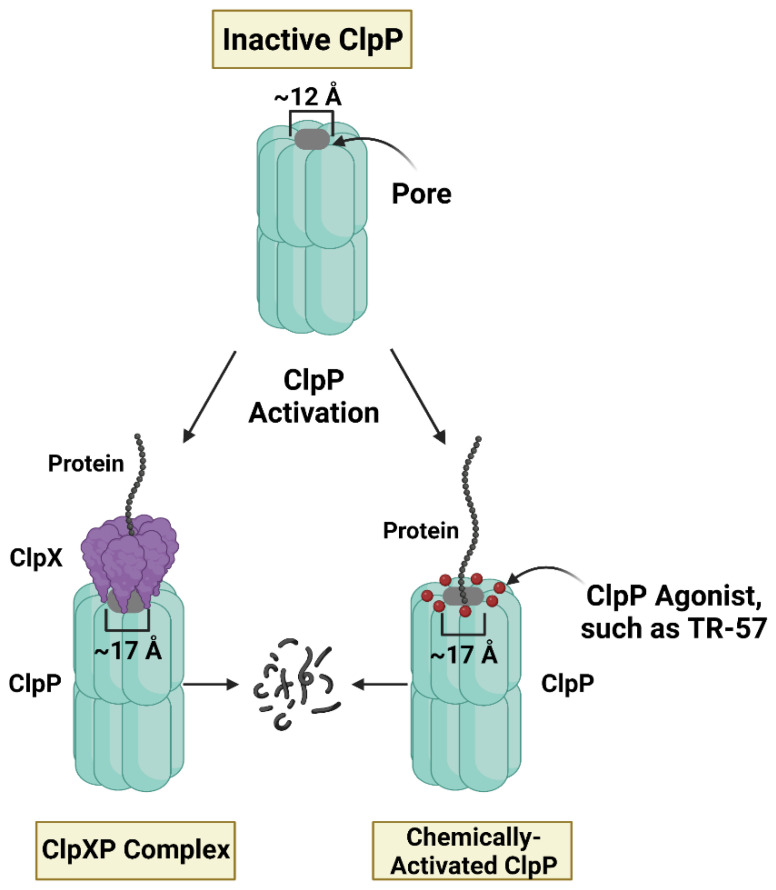
**Regulated and unregulated activity of mitochondrial caseinolytic protease ClpP.** The above figure depicts methods of ClpP activation. The protease ClpP is composed of 14 ClpP subunits, assembled into 2 heptameric rings. ClpP is typically bound to ClpX, an ATP-dependent unfoldase that contributes to ClpP proteolytic specificity. ClpP agonists bind to ClpP in the same location as ClpX, causing the barrel-shaped pore in ClpP to open. The closed pore is ~12 Å and the opened pore is ~17 Å, as determined by Ishizawa et al. [68]. With the pore open, ClpP is active and can degrade proteins, so ClpP agonists are able to increase proteolysis in the absence of ClpX.

**Figure 3 biomedicines-11-01598-f003:**
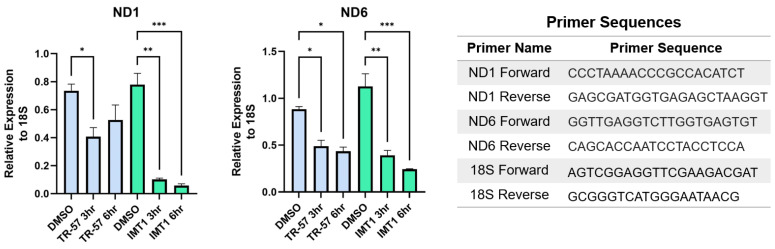
**TR-57 and IMT1 both inhibit mitochondrial transcription from the heavy and light strand**. HEK293T cells were plated at 4 × 10^5^ cells/well in 6 well plates, allowed to adhere overnight, and treated with TR-57 (150 nM) and IMT1 (10 µM) for indicated times. Cells were washed 3 times with cold DPBS, scraped, and pelleted. RNA was isolated using the RNeasy Plus Mini Kit (Qiagen) and residual DNA was removed using the RNase-Free DNase Set (Qiagen). cDNA was synthesized from 2 μg of extracted RNA in 20 μL total volume using the High-Capacity cDNA Reverse Transcription Kit (Applied Biosystems) and the Bio-Rad T100 thermal cycler. Quantitative PCR was performed using iTaq Universal SYBR Green Supermix (Bio-Rad). The nuclear reference gene 18S, mitochondrial heavy strand gene, ND1, and mitochondrial light strand gene, ND6, were quantified using the CFX96 Touch Real-Time System. A one-way ANOVA analysis was performed to determine significance; *p*-value < 0.05 (*), 0.01 (**), 0.001 (***). N = 2 biological replicates.

**Figure 4 biomedicines-11-01598-f004:**
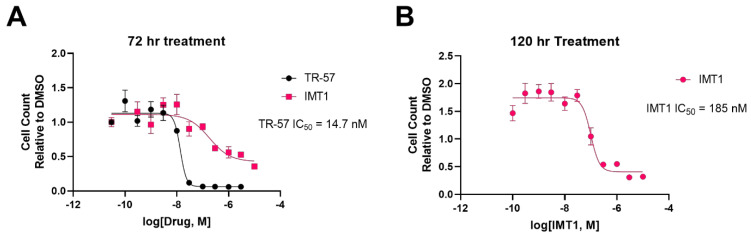
**The ClpP agonist TR-57 inhibits cell proliferation more potently and rapidly than the POLRMT inhibitor IMT1.** HEK293T cells were cultured in DMEM media supplemented with 10% fetal bovine serum and 1% antibiotic/antimycotic (Gibco). Cells were either plated in 96 well plates at 5 × 10^3^ cells/well, allowed to adhere overnight, and treated for 72 h with indicated concentrations of TR-57 and IMT1, or cells were plated at 1.5 × 10^3^ cells/well, allowed to adhere overnight, and treated with indicated concentrations of IMT1 for 120 h. After drug treatment for 72 (**A**) or 120 h (**B**), cells were stained with 2.5 µg/mL of Hoechst 33,342 and incubated for 20 min at 37 °C and 5% CO_2_. Total cell number was counted using the Celígo S imager (Nexcelom Biosciences). N = 2 biological replicates. The reported IC_50_ values are averaged from the 2 replicates.

**Figure 5 biomedicines-11-01598-f005:**
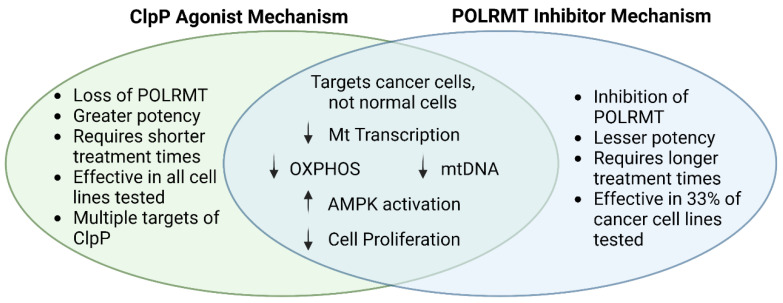
**Summary of similarities and differences between the mechanism of action of ClpP agonists and POLRMT inhibitors.** The Venn diagram represents observed mechanistic similarities and differences between ClpP agonists and POLRMT inhibitors. ClpP agonists and POLRMT inhibitors target mitochondria, impacting important mitochondrial functions, as well as inducing wide-spread cellular changes leading to inhibition of cancer cell proliferation. The common effects of these compounds on mtDNA and associated processes suggest that this may be important for the growth inhibition observed in cancer models.

**Figure 6 biomedicines-11-01598-f006:**
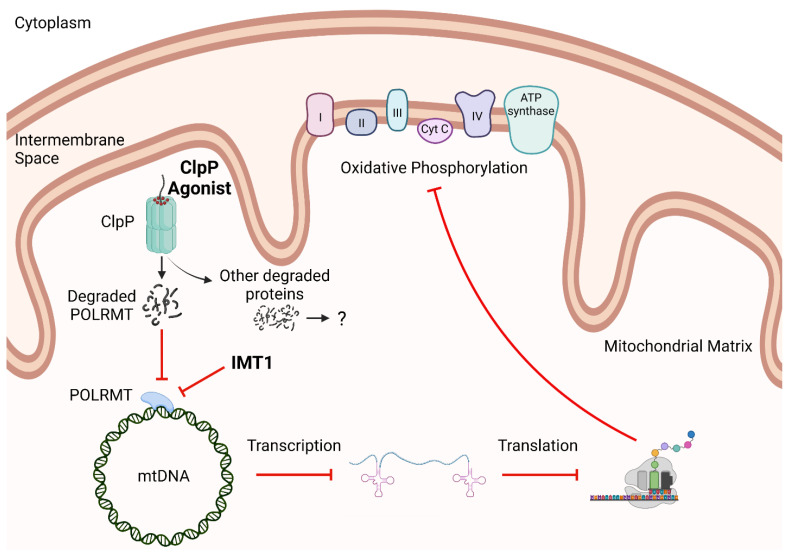
**Hypothesized mechanism of cell proliferation inhibition by ClpP agonists and POLRMT inhibitors.** The above schematic describes the loss of POLRMT activity, either by inhibition with IMT1, or protein degradation with ClpP agonists and subsequent effects on mitochondrial transcription. When transcription is inhibited, other key mitochondrial processes are impaired. Without transcription and translation of mitochondrially-encoded proteins, OXPHOS is inhibited which significantly affects cancer cell metabolism. The ability of ClpP agonists and POLRMT inhibitors to inhibit cancer cell proliferation is potentially due to this proposed mechanism of impairing mitochondrial transcription.

**Table 1 biomedicines-11-01598-t001:** Comparison of Potency in a Representation of Cell Lines Tested with POLRMT Inhibitors and ClpP Agonists.

Tissue	Cell Line	IMT1 IC_50_ (µM) *	ClpP Agonist	ClpP Agonist IC_50_ (µM)	Source
Bladder	J82	>30	ONC201	2.45	[70]
Breast	MDAMB436	0.56	ONC201	3.94	[70]
	MDAMB468	0.13	ONC201	1–6.3	[39,41]
	MDAMB231	>30	ONC201	3.0–7.0	[41,70,71]
			IMP075	10.1	[71]
			ONC212	<0.5	[103]
			ONC206	<1	[103]
			TR-57	0.017–0.0193	[35,78]
			TR-107	0.023–0.0294	[34,35]
Colon	CACO2	0.15	ONC201	15.5	[71]
			IMP075	2.5	[71]
	HT29	1.22	ONC201	5.07	[70]
	SW620	0.22	ONC201	9.8	[71]
			IMP075	0.6	[71]
Kidney	HEK293T	0.185	TR-57	0.0147	Daglish, et al.
Liver	HEPG2	0.12	ONC201	6.2–12.4	[39,71]
			ONC212	<1	[103]
			ONC206	<1	[103]
			IMP075	3.1	[71]
Lung	A549	0.64	ONC201	9	[71]
			IMP075	1.6	[71]
Ovary	IGROV1	>30	ONC201	5.55	[70]
	OVCAR3	2.23	ONC201	1.98	[70]
	SKOV3	>30	ONC201	2.3–20.5	[70,71]
			IMP075	4.1	[71]
Pancreas	BXPC3	0.27	ONC201	12.5	[72]
			ONC212	0.25	[72]
	PANC1	0.79	ONC201	3.13	[72]
			ONC212	0.17	[72]
Prostate	22RV1	0.82	ONC201	1.0–2.5	[70,72]
	DU145	>30	ONC201	>10	[72]
	PC3	>30	ONC201	2.2–20.1	[39,70,72]
Skin	A375	>30	ONC201	12.2	[71]
			IMP075	5	[71]

* All IMT1 IC_50_’s come from [20]. Ranges indicate >2 different IC_50_ values.

## Data Availability

Data will be made available upon request.

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
