# Peer review of "Targeting Mitochondrial DNA Transcription by POLRMT Inhibition or Depletion as a Potential Strategy for Cancer Treatment"

_biomedicines, 2023, doi:10.3390/biomedicines11061598_

Round 1

Reviewer 1 Report

There is an increased research interest to discover new inhibitors of the mitochondrial DNA-dependent RNA polymerase (POLRMT) by virtue of potential anti-cancer activity. Interestingly, recent studies have found that small molecule regulators of the mitochondrial matrix protease ClpP are potential inhibitors of mitochondrial transcription, triggering the rapid depletion of POLRMT and other proteins, resulting in inhibition of mitochondrial transcription and growth arrest.

In this Review, Daglish et al., compared the potential anti-cancer effects of these different inhibitors.

The Review is interesting but it is not always clear, well organized and discussed. Concepts are lacking of proper background and references need to be improved.

Minor comments:

-       Before introducing abbreviations please be careful to open them.

-       All References come after punctuation throughout the whole Review. Please, adjust it.

-       It would be great if the authors could contextualize a little bit more the results introduced at the end of section 5.5. “Inhibition of mitochondrial transcription”. The previous paragraph is discussing about the need of additional studies of CIpP agonist in the context of cancer cells, so these results, even though interesting, come from the blue.

-       First paragraph of section 6.1. “Differences in treatment response times”: please, provide references.

-       Figure 4: The authors should “justify” the use of this data. If I look at Table 1, what the authors want to add to this summary Table with their results? As above, in section 5.5, the authors should contextualize a little bit more the results presented.

-       Figure 5: “Targets cancer cells, not normal cells”. Could the authors justify this statement? In the present Review, the authors presented data using HEK293T cells that somehow contradicts this statement.

-       “Summary” section: “Importantly ClpP agonists and POLRMT inhibitors inhibit cancer cell proliferation with minimal effects on normal cells”. Please, provide relative references.

English is fine.

Reviewer 2 Report

This manuscript provides a detailed investigation into the mechanistic similarities and differences between ClpP agonists and POLRMT inhibitors, two drug classes that have shown promise as anti-cancer therapeutic strategies. The work carried out is thorough and the results are presented clearly, with the conclusion that both drug classes act by interrupting mitochondrial transcription and depleting mitochondrial DNA (mtDNA), yet they demonstrate significant differences in their effectiveness and resistance mechanisms.

The study demonstrates that both ClpP agonists and POLRMT inhibitors disrupt mitochondrial transcription and deplete mtDNA, which leads to the inhibition of cancer cell proliferation.The paper highlights distinct differences between the two drug classes, including their potency, response times, and effectiveness against different cancer cell types.This work raises important questions and hypotheses about why these drugs inhibit cancer cell growth but not normal cell growth, suggesting further research directions.The research is comprehensive and systematically conducted. The experimental design is sound and the methods used are appropriate.The manuscript clearly presents the results and discussions, making it easy for readers to follow the logic and understand the outcomes.The paper provides novel insights into the mechanism of action of both drug classes, which could pave the way for further research and potential therapeutic applications.

While the paper hypothesizes why these drugs inhibit cancer cell growth but not normal cell growth, more empirical evidence or experimental data would strengthen this claim.Although the authors have provided a comprehensive analysis of the response of different cancer cell types to ClpP agonists and POLRMT inhibitors, the underlying reasons for these differences are not thoroughly explored.The study lacks a detailed discussion on the possible side effects of these drugs. While it is mentioned that ClpP agonists have minimal side effects, an in-depth analysis would be beneficial.

The authors should consider including more empirical data or experimental evidence to back up their hypothesis regarding the selective inhibition of cancer cell growth.

A more detailed investigation into the reasons behind the differences in response across various cancer cell types would provide a deeper understanding of the mechanisms of action of these drugs.

An in-depth discussion on the potential side effects of these drugs would enrich the manuscript and provide a more balanced perspective.

Overall, this manuscript presents an important contribution to the understanding of the mechanisms of action of ClpP agonists and POLRMT inhibitors. With some minor revisions, it has the potential to provide valuable insights for cancer therapeutics research.

Round 2

Reviewer 1 Report

The Review has improved.

Minor comments:

-Figure 3 legend: N=2; 2 biological replicates in duplicate, triplicate...? Please, specify it.

-Figure 4 legend: same comment of Figure 3 legend.

-Figure 4: Please, check X axis. What does it mean -12, -10, -8...? It is a little bit confusing if referring to drug concentration. Please, adjust it.

-Table 1: Kidney_Source: Instead of Figure 4, I would refer to "Daglish et al.".

Minor English editing is required.

Author Response

The Review has improved.

Minor comments:

-Figure 3 legend: N=2; 2 biological replicates in duplicate, triplicate...? Please, specify it.

-Figure 4 legend: same comment of Figure 3 legend.

Thank you for your thorough evaluation of our paper in this second round of comments. The legends for Figure 3 and 4 have been updated to say N=2 biological replicates to indicate that the data is representing two biological replicates from each experiment. For Figure 3, the experiments were run with two technical replicates. For Figure 4 the experiments were run with four technical replicates.

-Figure 4: Please, check X axis. What does it mean -12, -10, -8...? It is a little bit confusing if referring to drug concentration. Please, adjust it.

The legend for Figure 4 was also updated to show log[Drug, M] to specify the drug concentration. Thank you for catching this mistake!

-Table 1: Kidney_Source: Instead of Figure 4, I would refer to "Daglish et al.".

Table 1 was updated to refer to “Daglish, et al.” instead of Figure 4 as suggested.

Also, since the reviewer suggested “Minor English editing”, we had a fluent English speaker read the manuscript and made changes based on his suggestions on lines 32, 100, 131, 136, and 291.

Reviewer 2 Report

I appreciate the authors' response to my feedback. All of my comments and concerns have been effectively addressed. However, I would like to suggest a small modification regarding the figures. Would it be possible to standardize the font used in the figures to either Times New Roman or Arial for consistency?

Author Response

I appreciate the authors' response to my feedback. All of my comments and concerns have been effectively addressed. However, I would like to suggest a small modification regarding the figures. Would it be possible to standardize the font used in the figures to either Times New Roman or Arial for consistency?

We would like to thank the reviewer for their kind comments on our manuscript. As requested, we have changed the font on all the figures to Arial so that they are all consistent. Thank you for this suggestion!